# Control of Synapse Structure and Function by Actin and Its Regulators

**DOI:** 10.3390/cells11040603

**Published:** 2022-02-09

**Authors:** Juliana E. Gentile, Melissa G. Carrizales, Anthony J. Koleske

**Affiliations:** 1Stanley Center for Psychiatric Research, Broad Institute of MIT and Harvard, Cambridge, MA 02142, USA; 2Department of Molecular Biophysics and Biochemistry, Yale University, New Haven, CT 06520, USA; melissa.carrizales@yale.edu; 3Department of Neuroscience, Yale University, New Haven, CT 06520, USA

**Keywords:** actin, synapse, neurotransmission, presynaptic, postsynaptic, electron microscopy, endocytosis, vesicle recycling

## Abstract

Neurons transmit and receive information at specialized junctions called synapses. Excitatory synapses form at the junction between a presynaptic axon terminal and a postsynaptic dendritic spine. Supporting the shape and function of these junctions is a complex network of actin filaments and its regulators. Advances in microscopic techniques have enabled studies of the organization of actin at synapses and its dynamic regulation. In addition to highlighting recent advances in the field, we will provide a brief historical perspective of the understanding of synaptic actin at the synapse. We will also highlight key neuronal functions regulated by actin, including organization of proteins in the pre- and post- synaptic compartments and endocytosis of ion channels. We review the evidence that synapses contain distinct actin pools that differ in their localization and dynamic behaviors and discuss key functions for these actin pools. Finally, whole exome sequencing of humans with neurodevelopmental and psychiatric disorders has identified synaptic actin regulators as key disease risk genes. We briefly summarize how genetic variants in these genes impact neurotransmission via their impact on synaptic actin.

## 1. Introduction

The neuronal synapse is the elemental component of communication in the nervous system and is composed of both pre- and post-synaptic compartments. The shape, size, alignment, and organization of these compartments can change dramatically in response to input and activity patterns, and these changes govern the probability and magnitude of synaptic transmission. Both the pre- and post-synaptic specializations are enriched with the cytoskeletal protein, actin. Actin filament networks provide the shape and size to these compartments, organize presynaptic neurotransmitter release and post-synaptic receptors, and play vital roles in trafficking key molecules into and out of synapses. The polymerization, organization, stability, and turnover of actin filaments is regulated by a key set of actin regulators, whose activities are controlled by both developmental and synaptic-activity-based mechanisms. Not surprisingly, mutations in key actin regulators are widely associated with brain disorders. We discuss the various roles played by actin at the synapse, how these functions are regulated, and how disruption of key actin regulatory mechanisms contributes to human brain disorders.

## 2. Discovery and Study of Actin at the Synapse

At the turn of the 1900s, Santiago Ramón y Cajal laid the basis for modern-day neuroscience in his earliest visualization and artistic renderings of pyramidal neurons. He made some of the earliest claims that neurons are individual units of the nervous system and are covered by small protrusions he referred to as spines, now referred to as dendritic spines [1]. Since this pioneering work, many new technologies have expanded on the methods used by Ramón y Cajal, and have advanced the way we now understand neurons, dendritic spines, and the way individual neurons communicate across synapses. We have summarized this, as well as other advancements described in this section, in Figure 1.

The development of better fixation and sample processing techniques coupled with the use of electron microscopy (EM) were instrumental in visualizing and characterizing the cytoskeleton [2]. Microtubules were easily visualized throughout dendrites and axons due to their inherently bigger size. Actin filaments, termed microfilaments in the early literature, were first shown to be important for axon elongation and microspike formation on growth cones emanating from the axon [3]. Still, visualization via thin section EM made it difficult to observe the detailed ultrastructure of actin filaments. Labeling actin filaments with the S-1 fragment of myosin enabled visualization more readily [4,5], both in thin section preparations and in whole mount transmission EM. In addition to visualizing actin filaments in growth cones, Markham and colleagues showed that dendritic spines contained a network of filaments that increase in number and complexity as the spines mature.

Later, several groups identified EM microfilaments as actin filaments through anti-actin immunoreactivity and rhodamine phalloidin staining. In the early 1940s, Coons and colleagues first demonstrated that tissue could be immunolabelled with fluorescent antibodies for visualization in light microscopy [6]. However, it took several decades before this approach was employed to visualize the neuronal cytoskeleton. In fact, this approach was first used not in neurons, but in neuroblastoma cell lines as a model to understand the concentration and dynamics of cytoskeletal components in a model of the developing nervous system [7]. The presence of actin at synapses was first deduced by the immunoreactivity for actin in synapse-rich brain regions and in purified synaptic preparations from tissue—so-called synaptosomes [8]. This was consistent with reports of filamentous “flocculent” material found to be enriched in electron micrographs of dendritic spines [9]. Immunostaining of tissue in tandem with electron microscopy revealed especially high concentrations of actin in dendritic spines [10,11], although actin filaments were also observed presynaptically, in association with synaptic vesicles and short filaments thought to represent synapsins [12,13]. Incubation of hippocampal tissue with the S-1 myosin subfragment followed by electron microscopy showed that actin filament barbed ends were closely associated with the plasma membrane and postsynaptic density (PSD) of dendritic spines [10,14]. These approaches also revealed that actin filaments contacting the PSD were interconnected via elaborate networks with filaments associated with the spine apparatus, a form of endoplasmic reticulum unique to dendritic spines. This suggests that filamentous actin arrays could communicate events occurring at the membrane with the underlying synaptic ultrastructure [12].

By the end of the 1990s, fixation and staining protocols were refined to their contemporary state and probes such as fluorescent phalloidin, anti-tubulin, and MAP2 antibodies became standard neuronal labels [15,16]. Confocal microscopes and their capability to obtain optical sections enabled reconstruction of the cytoskeleton in collateral axon branches [17]. New image processing techniques, in conjunction with microscopes equipped with cooled CCD cameras, were now able to image two different targets almost simultaneously. These advancements allowed for the identification of the mechanistic relationship between the actin cytoskeleton and mitochondrial transport, where mitochondria accumulate at areas of actin polymerization and depolymerization, consistent with high energy consumption at these sites [18]. Platinum replica electron microscopy, as refined by Svitkina and colleagues, enabled direct visualization of the actin ultrastructure in dendritic spines and presynaptic boutons [19]. This approach used rapid detergent treatment or physical disruption to remove the cell membrane while maintaining preservation of the fragile actin ultrastructure. Samples were then stained with the heavy metal platinum to create a replica of the cytoskeleton, which can be imaged by a transmission electron microscope [20]. The use of this technique allowed the observation of an interconnected actin network in the spine head, spine necks, and dendritic filopodia, that was highly branched. In conjunction with simultaneous antibody labeling, researchers could see the network was interwoven with actin regulators such as the Arp2/3 complex, capping protein, and myosin II. While this study revealed the architecture of the actin network and actin-associated proteins underlying the synapse, the membrane extraction processes likely remove fine membrane-associated peripheral structures.

In the past few decades, use of fluorescent protein-labeled actin (e.g., GFP-actin) enabled direct visualization of the dynamic evolution of actin structure in cells using live-cell microscopy. Fluorescent protein-tagged actin incorporates into the actin network, yielding staining patterns that are indiscernible from anti-actin staining. Since the creation of fluorescent phalloidin and GFP-actin, many additional strategies have been used to label actin structures in live and fixed cells, tissues, and whole organisms. Many probes have been created by fusing a fluorescent protein to the actin-binding domain of a known actin-binding protein. One example is Lifeact, a probe for filamentous actin in which the first 17 amino acids of the yeast actin-binding protein Abp140 are fused to GFP [21]. Although fluorescent derivatives of phalloidin are generally thought to provide the most complete picture of the filamentous actin cytoskeleton, Lifeact has remained widely used due to its small size, lack of impact on actin polymerization/depolymerization rates, and lack of competition with homologous proteins in higher eukaryotes [21,22]. Finally, the use of photoactivatable GFP-actin fusion (e.g., paGFP-actin) has enabled photon-induced labeling of bulk or single actin molecules so that their fates can be monitored. This labeling approach showed that actin polymerization rates were higher at the spine edge than at the spine base and that actin polymers undergo retrograde flow from the periphery toward the spine base [23]. Live tracking of labeled actin also revealed pools of actin in spines that undergo dramatic expansion in response to stimulation [23]. The application of actin polymerization inhibitors rapidly arrested spine motility, indicating that actin filament networks are a primary determinant of spine shape and structural plasticity and that actin polymerization is indeed needed for spine expansion [24].

Along with better imaging probes to label actin, advanced techniques have increased optical resolution to image actin and other regulatory proteins with incredible spatial accuracy, well below the ~250 nm diffraction limit of optical microscopy. Photo-activated localization microscopy (PALM) and stochastic optical reconstruction microscopy (STORM) push resolution limits by sparsely activating fluorescent molecules, which form diffraction limited spots [25,26]. Each molecule can be localized to much higher precision by using mathematical models to reconstruct a subdiffraction image of single molecule positions. To avoid the accumulation of activated fluorophores, PALM exploits the spontaneously occurring phenomenon of photobleaching, while STORM takes advantage of reversible switching between a fluorescent on-state and its dark off-state. PALM and photo-activatable actin probes have been used to measure the trajectories of single actin molecules incorporated into filaments, revealing highly heterogeneous flow dynamics of actin in spines [27], with actin that is newly incorporated into actin filaments near the PSD moving faster than those incorporating into filaments elsewhere in the spine [28]. In stimulated emission depletion (STED) microscopy, the focal spot of excited light is overlapped with a doughnut-shaped spot of lower photon energy light, quenching excited molecules in the excitation spot periphery and resulting in a subdiffraction-sized fluorescent spot [29]. Due to the nature of stimulated depletion, imaging probes must be both photostable and bright. Silicone-rhodamine desbromo-desmethyl-jasplakinolide (SiR-actin) was created as an alternative to Lifeact for these approaches and has since been used in live imaging STED to visualize the periodic actin structures in dendritic spine necks (see Section 4.1) [30,31,32]. These studies provided some of the earliest evidence that the actin cytoskeleton may underlie functional and structural changes of the synapse.

While they are powerful tools, super-resolution nanoscopy approaches require specialized microscopes and sophisticated data analysis that may not be accessible for every lab. A more tractable and accessible technique for future lines of research on the actin cytoskeleton and its regulatory proteins may be expansion microscopy [33,34], which allows for the visualization of small, typically diffraction limited, structures by physical magnification. The basis of this method is to polymerize an acrylamide gel within a cell or tissue, and crosslinking endogenous proteins to that gel. Next, protein digestion or SDS denaturation is used to ensure isotropic expansion when the proteins are separated during expansion of the hydrogel. This expanded gel can then be stained for different targets and imaged via conventional confocal microscopy. Expansion microscopy yields exceptional resolution in brain tissue (~70 nm) [33] and can recapitulate diffraction limited structures that had only been seen previously using super-resolution techniques [35]. While expansion microscopy presents its own challenges, it should allow the field to image actin ultrastructure, its associated cofactors and regulators, and their relationship to overall synaptic structure using less damaging sample preparation techniques. From the earliest Ramón y Cajal drawings of a neuron, the field has made tremendous progress in understanding the structure and function of the actin network at the synapse. Our review will explore seminal findings and discuss ongoing and future challenges in the field.

## 3. Chemical Probes to Study Actin Function in Neurons

Chemical compounds that impact the actin polymerization cycle have been widely used to probe the roles of actin in synapse development, structure, plasticity, and function. In cells, actin filaments polymerize from a freely diffusing pool of actin monomers to form an elaborate network of actin filaments (Figure 2A). Actin monomers have 4 subdomains, which are split by a cleft that binds ATP. Many actin-binding proteins bind in the groove between subdomains 1 and 3 [36], which is still accessible for binding in filaments (Figure 2B). Actin monomers assemble into asymmetric filaments where ATP-actin monomers are preferentially added to the barbed end and ADP-actin monomers dissociate from the pointed end [37]. The difference in polymerization behavior at each filament end leads to turnover of the filament, a process known as treadmilling. Chemical compounds that bind and affect different aspects of the filament lifecycle have been used as probes to study actin behavior in cells. Figure 2C shows the approximate binding location of these chemicals on actin and Table 1 summarizes how these chemical probes impact actin dynamics.

## 4. Organization and Roles of Actin in Neuronal Processes and within the Synapse

Dynamic actin plays key roles in the developing neuron by powering the navigation of neuronal processes and formation of proper synaptic connections. Landmark papers in the field (see above) have revealed an enrichment of actin in presynaptic boutons of the axon and in postsynaptic dendritic spines. The importance of actin networks and their remodeling in spines continues even within mature synapses, which can undergo near constant activity-dependent remodeling to adapt their structure and function in response to altered input and activity patterns. In the following sections, we will review actin organization in the various compartments of a neuron, as well as key roles that actin has in vesicle release, endocytosis, and the organization of neurotransmitter receptors. We have also summarized key roles of actin in the various neuronal compartments in Table 2.

### 4.1. Actin in Axons and Dendrites

In the axon, actin plays critical roles in axon guidance and terminal arborization (reviewed in [42]). It also plays critical roles in organizing and regulating neurotransmitter release from presynaptic terminals. Using the powers of STORM and STED, actin filaments, along with adducin, form periodic ring-like structures around the circumference of the axon shaft, with a spacing of 180–190 nm between rings (Figure 3) [43,72]. Spectrin is the key for dictating the spacing of these rings, as the knockdown of β2-spectrin disrupts these regular periodic actin:spectrin structures in both axons and dendrites and perturbations even extend into the spine neck [32,73]. This periodic actin:spectrin structure is also found in dendrites and a regularly spaced actin:spectrin latticework is observed in the neuronal soma [72,73], making it a common cytoskeletal structure beneath the neuronal membrane. It has been proposed that the actin:spectrin framework provides mechanical support for the integrity of long-lived neuronal structures, a role similar to the one it plays in red blood cells [41]. Consistent with this hypothesis that the actin:spectrin framework provides mechanical support, axons spontaneously break in *C. elegans* neurons lacking spectrin [58]. The actin:spectrin framework and associated cytoskeletal proteins (e.g., adducin) also organize axonal sodium channels, which exhibit periodic localization that is in register with the periodicity of the actin:spectrin-based cytoskeleton [43]. Beneath this periodic, mechanical lattice, axons and dendrites also contain single and bundled filaments that lie parallel to the shaft axis (Figure 3) [41,73,74]. Recent data suggests that segments of dendrites are dominated by either periodic actin rings or longitudinal actin arrays and increasing neuronal activity might convert the actin rings to longitudinal structures, although the functional significance of these changes is not clear [78].

In addition to dendritic periodic lattices and parallel bundled filaments, there are highly branched actin networks that reside in the dendrite and sit at the base of dendritic spines [19]. One study showed that dynamic microtubules preferentially target spines that are undergoing actin remodeling in an activity-dependent manner [74]. Actin stabilizing drugs, such as latrunculin A, or knockdown of the important actin remodeling protein cortactin, resulted in significantly less frequent microtubule-spine targeting events. Direct interactions between the microtubule plus end-binding protein and actin regulators were not required for microtubule targeting, suggesting that actin may guide or push the growing microtubule tip into the spine.

### 4.2. Presynaptic Actin Regulates Synaptic Vesicle Release and Vesicle Endocytosis

Actin ultrastructure in the presynaptic bouton and dendritic spine are similar in overall design [19], both comprised of an intricate combined meshwork of short, branched and bundled filaments. Presynaptic actin forms a mesh around synaptic vesicles to act as a scaffold for synaptic vesicle regulators, such as synapsin and bassoon (Figure 4) [75,83]. Indeed, inhibitor studies have revealed diverse roles for actin in the synaptic vesicle cycle. For example, treatment of neurons with the actin polymerization inhibitor latrunculin A transiently increases mini-excitatory postsynaptic current (mEPSC) frequency, a proxy for single synaptic vesicle fusion events, while not altering mEPSC amplitude. This observation suggests that actin negatively regulates synaptic vesicle release probability by providing a tether or physical barrier to restrain vesicle fusion at the active zone [83]. These findings were further corroborated with a microscopy-based approach showing that the rate of presynaptic vesicle fusion was 20% faster in the presence of latrunculin A [75]. Latrunculin A does not detectably perturb synaptic vesicle clustering or presynaptic levels of synaptic vesicle protein 2 or synaptophysin, at least at the light microscopy level, suggesting that gross organization of presynaptic release sites does not require ongoing actin polymerization [16]. In contrast, one study showed that long-term latrunculin A treatment reduced mESPC frequency, suggesting that some aspect of the release machinery may be sensitive to actin depolymerization [79]. One possibility is that actin polymerization is required for the endocytosis of vesicles to fuel synaptic vesicle replenishment (see next two paragraphs), which would become depleted in long-term latrunculin A treatment.

Beyond its roles in regulating neurotransmitter release, others have identified roles for actin in endocytosis and the transport of recycled vesicles to the synaptic vesicle cluster (Figure 4) [76,84]. Actin:β2-spectrin clusters form a laddered ring appearance in the frog neuromuscular junction (NMJ), where they surround synaptic vesicle clusters at release sites [85]. At both the NMJ and reticulospinal synapses, actin filaments are concentrated at endocytic zones that lie lateral to vesicle release sites, where they are tightly associated with presynaptic endocytic intermediates in electron micrographs of synaptosomes [77]. Using the lamprey giant reticulospinal synapse, in conjunction with microinjections of compounds that target actin and thin section EM, Shupliakov and colleagues described membrane proximal actin cytomatrix lateral to the vesicle cluster [76,84]. Synaptic stimulation increased the density of filaments that were associated with both clathrin-coated pits and with synaptic vesicles. Following microinjection of these axons with the actin stabilizer phalloidin, synaptic vesicles appeared trapped along these proliferated filaments as they were being recycled [76]. Phalloidin injection also yielded an expansion of the presynaptic membrane and unusual elongated clathrin-coated endosomal structures [76]. The disruption of actin polymerization with latrunculin or swinholide inhibits vesicle endocytosis in a variety of model systems [80] leading to a significant increase in unconstricted clathrin-coated pits [81]. Collectively, these observations point to a direct role for actin in vesicle endocytosis.

Presynaptic vesicle endocytosis occurs via multiple distinct pathways in different model systems [82]. These pathways occur on different timescales, differentially contribute to replenishment of the readily releasable and reserve pools (RRP and RP) of vesicles, and require distinct molecular components, including their dependence on actin. At the fly neuromuscular junction (NMJ), robust stimuli will deplete vesicle pools, which are replaced by endocytosis. Recovery of the RRP is rapid, while recovery of the RP occurs on a longer time frame [39]. The disruption of actin with cytochalasin D impairs the RP recovery, while leaving the fast recovery of RRP largely intact [39]. Similar experiments have verified a requirement for actin in RP recovery at the frog NMJ [80] and at mouse hippocampal mossy fiber synapses [44]. Tetanic stimulation of the toad NMJ can trigger wholesale endocytosis of large membrane-bound compartments, which are recycled into vesicles following endocytosis [59]. While most studies have used drugs to probe the role of actin, Wu et al. tested how the knockout of the β and γ isoforms of actin directly impacts endocytosis. In an elegant series of experiments, they demonstrated that both actin isoforms were required for all forms (fast, slow, bulk) of endocytosis at the Calyx of Held and for both slow and bulk endocytosis at hippocampal synapses [45]. The defects in β-actin-deficient synapses could be rescued by re-expression of β-actin, but not a polymerization-defective mutant. The depletion of β-actin also significantly reduced the frequency and constriction of endocytic invaginations, strongly suggesting that actin polymerization promotes endocytic vesicle constriction. This constriction may also require formins and myosin II isoforms, inhibitors that slow endocytosis and arrest membrane invaginations at an uncoated and unconstricted state [46]. Thus, it appears that actin filaments play vital roles in the most critical aspects of presynaptic function in the active zone: actin filaments gate synaptic vesicle release and possibly organize vesicles for optimal release and actin polymerization aids in endocytosis and vesicle recycling.

### 4.3. Actin Regulates Changes in Neuronal Shape

In addition to its role in presynaptic neuronal function, actin is important in regulating the shape of the neuron. Axons extend to the proper brain region and synaptic targets, driven by actin polymerization in the growth cone. External guidance cues signal through growth cone receptors and trigger signaling cascades that regulate actin-binding proteins to locally modulate actin polymerization (reviewed in [47]). In the developing brain, actin expression increases rapidly before the development of nascent spines [48], suggesting a role for actin in spinogenesis and in driving the final structure of the spine (Figure 5).

Most dendritic spines in the adult brain consist of an expanded head connected to the dendrite by a narrow neck, often called mushroom spines, but a variety of other spine shapes also exist (see [86] for a review on spine shape and their dynamics). Stubby spines are characterized by a lack of neck, while filipodial spines lack a bulbous head; both are common during the early stages of postnatal development (Figure 5). More recently, other groups have described newer classes of spine structures such as branched and cup spines [49]. The actin cytoskeleton is the principal component supporting spine shape and stability [24,60,61,87,88].

The plasma membrane of the spine neck is decorated with periodic actin rings [32], while internally the spine neck is supported by both longitudinal and branched actin networks. The spine head is enriched in branched filament networks oriented with plus ends apposed to the membrane [19]. This spine actin network interfaces with cell adhesion molecules that link to the extracellular matrix and pre-synaptic compartment, to coordinate spine shape changes and stability [89].

Actin dynamics drive structural changes in the spine head and neck, which can alter synaptic function. Parallel bundled filaments and spectrin:actin rings constrain the spine neck to create a physical [32,62] and electrochemical barrier that compartmentalizes the spine head from the dendrite. Indeed, voltage pulses propagating either to or from the spine are attenuated in proportion to the spine neck length [50,51], regardless of the location or size of the spine. While both the inner longitudinal filaments and spectrin:actin rings are important for the integrity and shape of the spine neck, the inner filaments act as the diffusion barrier to molecules moving into and out of the spine, while the external periodic rings function as the electrochemical barrier to signal propagation, similar to their role in axons [90].

As spines grow and shrink in development, or in response to synaptic activity, shape changes are driven by changes in the underlying actin cytoskeleton. Under basal neuronal activity, the amplitude of synaptic currents scales with spine head volume, as revealed by measuring synaptic responses to two-photon glutamate uncaging at single synapses [91,92]. The pattern of spine activation (e.g., low- vs. high-frequency stimulation) evokes changes in spine shape, mediated by actin cytoskeleton remodeling. High-frequency stimulation that causes long-term potentiation (LTP), which is believed to be a proxy for memory formation, promotes spine head enlargement [71,93]. These same stimulations lead to a long-lasting increase in the stable actin pool [23,65]. Not only do spine heads increase in size following these potentiation events, but novel super-resolution imaging approaches revealed that spine necks also increase in diameter [66], an event that may also be driven by the expanding actin network. Conversely, low-frequency stimulation that causes long-term depression results in spine head shrinkage or spine loss in already small spines, and net destabilization of the actin network [93].

### 4.4. Postsynaptic Actin Organizes Neurotransmitter Receptor Machinery at Excitatory and Inhibitory Synapses

Post-synaptic neurotransmitter receptors, adhesion molecules, and associated scaffolding proteins concentrate in the protein-rich postsynaptic density (PSD) of mature spines. Post-synaptic actin serves both as a tether and a platform on which receptors traffic into and out of the synapse (Figure 6) [67]. Electron microscopy revealed that actin filaments extend into the PSD [10,11,14,94], where they can function to tether or concentrate postsynaptic scaffolding proteins or neurotransmitter receptors in this region or interact directly with the receptors to regulate their gating properties (reviewed in [95]). Excitatory neurons contain two classes of ionotropic glutamate receptors, α-amino-3-hydroxy-5-methyl-4-isoxazolepropionic acid receptors (AMPA receptors, AMPARs) and N-methyl-D-aspartate receptors (NMDA receptors, NMDARs), which exhibit different dependencies on actin for their clustering at synapses. The treatment of cultured hippocampal pyramidal neurons with latrunculin A leads to a 40% reduction of spines containing AMPA receptor clusters [16]. This loss of AMPA receptors may reflect a requirement for dynamic actin remodeling for sustained AMPAR trafficking to or clustering within synapses and is consistent with a greater lateral exchange rate of AMPARs between synaptic and extrasynaptic space in comparison to NMDARs [94]. Further corroborating these findings, latrunculin A treatment increased single AMPAR diffusion in both synapses and extrasynaptic spaces, indicating that actin somehow restricts AMPAR mobility [96]. Actin also serves as a scaffold for myosin Vb to traffic AMPARs from recycling endosomes into the PSD during activity-based plasticity [97]. The actin-severing protein cofilin is also critical to activity-dependent increases in synaptic AMPAR levels, possibly acting to locally remodel actin to facilitate insertion [98]. Together, these studies suggest that dynamic actin filaments are required for proper insertion of AMPARs at the synapse, where actin continues to provide a means to promote their clustering.

In a notable contrast, NMDARs remain clustered following latrunculin treatment, but these clusters become delocalized from their presynaptic contacts [16]. Notably α-actinin2, which binds both actin filaments and the cytoplasmic tails of several NMDA receptor subunits, is also dispersed from synapses by latrunculin treatment, strongly suggesting that actin tethers NMDA receptors within the synapse via a network of protein:protein interactions. Intriguingly, the NMDAR GluN2B subunit cytoplasmic tail binds the adaptor protein Nck2 [99]. Nck2 is known to recruit N-WASp to activate Arp2/3 complex-mediated actin filament nucleation (Figure 7A). This observation raises the possibility that NMDARs serve as a platform from which to promote actin polymerization. Indeed, significant remodeling of PSD scaffolding proteins, observed by live imaging, is enhanced by activity and can be arrested by latrunculin treatment [100]. In fact, key actin nucleation regulators, including the Arp2/3 complex and WAVE complex components, are concentrated at nanodomains just adjacent to the PSD and these are hotspots for new actin polymerization [28,68]. Hence, activation of NMDARs might trigger adjacent actin polymerization to remodel the PSD and possibly alter its nanoscale alignment with the presynaptic release machinery, a model initially proposed by Blanpied and colleagues.

Actin also plays vital roles in postsynaptic organization at inhibitory synapses, although its specific roles may differ, depending on cell type and context. Clusters of glycine receptors (Gly receptors, GlyRs) at inhibitory synapses on spinal cord neurons seem to depend partly on microtubules (MTs), as receptor clustering is greatly reduced by treatment with MT inhibitors [101]. GlyRs bind to gephyrin, which in turn binds to MTs, may serve as the cytoskeletal anchor for GlyRs. In fact, treatments with MT inhibitors causes a loss of the inhibitory scaffolding protein gephyrin from synapses. Interestingly, in static imaging, reduction of actin by latrunculin A reduced GlyR receptor cluster size but increased their packing density, consistent with the role of actin in restricting the spacing of GlyR placement at the synapse. Single molecule tracking of GlyRs have revealed a more nuanced role for actin in controlling synaptic GlyR levels (Figure 7B). Single synaptic GlyRs exhibit high mobility in extrasynaptic space and a more confined diffusion in synapses, but can exchange readily between extrasynaptic and synaptic areas via lateral diffusion. Disruption of actin with latrunculin expanded this synaptic diffusion area, increased exchanges of GlyRs into and outside of synaptic zones, and reduced their dwell time within the synapse [102]. These factors led to a net decrease of GlyRs in synapses following latrunculin treatment. Together, these data indicate that postsynaptic actin networks act as a dynamic tether to modulate GlyR levels and overall synaptic tone. In contrast, the clustering of gamma-aminobutyric acid receptors (GABARs) at inhibitory synapses is not disrupted by treatments that disrupt actin or MTs [15], suggesting that diverse mechanisms may tether inhibitory receptors in different neuronal subtypes.

Similar to its presynaptic roles, actin and actin-associated proteins play key functions in postsynaptic endocytosis. Endocytic structures can be visualized in lateral domains of the spine, away from the PSD, where key regulators of endocytosis (e.g., clathrin, dynamin, and AP-2) are enriched, as determined by immunogold electron microscopy [103]. Tonic glutamate stimulation of synapses yields endocytosis of AMPARs, which is blocked by jasplakinolide (Figure 7C) [104]. Actin-based motor Myosin 6 (Myo6) can associate in complexes with AMPARs and neurons deficient in Myo6 do not undergo AMPAR internalization after glutamate stimulation [52]. Myo6 is unique among myosins in that it tracks toward the actin minus ends, which are pointed away from the PSD, providing an elegant means to carry AMPAR-containing vesicles away from the synapse. It is worth noting that the peaks of actin assembly do not colocalize with concentrations of endocytic marker clathrin in the dendritic spine. It may be that some forms of post-synaptic clathrin-dependent endocytosis in spines do not require actin, or the pool of actin involved in endocytosis is just smaller and/or it turns over more slowly than the more concentrated dynamic actin pool at the PSD [28].

## 5. Different Actin Filament Pools in the Spine: Evidence, Localization, and Possible Functions

Dynamic actin rearrangements underlie morphological changes that regulate neuronal morphogenesis and are required for changes in synapse size and shape, neurotransmitter release, synaptic vesicle recycling, and synaptic neurotransmitter receptor number. However, to maintain the complex morphology of neurons and synapses, a minor pool of actin filaments in the spine are relatively stable and turn over in the order of tens of minutes [23]. These filaments are usually capped or bound along their lattice to prevent polymerization or depolymerization [105]. When activity-dependent growth cues signal the spine actin network to expand, an actin monomer pool feeds into the actin network and incorporates onto the barbed end of polymerizing filaments [23,53]. Filament polymerization is initiated by uncapping filament barbed ends or severing events that create new, uncapped barbed ends that are elongated [37]. In synapses, specifically, dynamic actin networks reside at the periphery of the spine and in association with the PSD [28,64,102], while the more stable actin filaments reside at the spine core (Figure 6) [23].

Actin-binding proteins bind and affect every stage of actin assembly: binding proteins that increase or decrease filament polymerization rates, nucleators that allow for nascent filament formation or branched filament formation from the side of a mother filament, severing proteins that break actin filaments, proteins that cap barbed or pointed ends to terminate elongation, and some even bind actin monomers to shuttle them to active areas of polymerization [37]. While experiments using purified proteins have been instrumental in assessing the roles of these proteins in actin polymerization, neuron experiments have been critical to understand how these activities contribute to the regulation of actin in dendritic spines and ultimately their effect on synapse shape and function. The geographic distribution of many actin regulators show remarkable compartmentalization in neurons [63] (summarized in Table 3 and depicted in Figure 6) that coincide with different actin pools. Seminal EM studies localized these proteins within three main compartments—the spine core at the spine center, the spinoplasm in a zone surrounding this core, and concentration in the electron-dense post-synaptic density (PSD), adjacent to or in contact with neurotransmitter receptors.

Although many actin regulators primarily reside in one subcompartment within the spine, some do undergo activity-dependent translocations. For example, profilin is targeted to spine heads upon activation of NMDARs, and this event coincides with a net reduction in dynamic spine shape changes [106]. Bosch et al. (2014) systematically examined the levels and spine localization of a panel of synaptic actin regulators following LTP induction [64]. They monitored single spines for 30 min following LTP induction and found many proteins (cofilin, actin, Arp2/3 complex, profilin, drebrin, Aip1, GluA1, α-actinin, CaMKIIα, and CaMKIIβ) were delivered to the spine. Interestingly, these proteins moved according to distinct dynamic patterns, in three sequential temporal phases. Many of the proteins translocated early into the spine were actin regulatory proteins responsible for actin severing, branching, and capping. This translocation of actin regulatory proteins caused a major reorganization of the actin network as soon as 20 s after the induction of LTP. Known stabilizers of actin filaments were transiently lost from the spine in this early phase but gradually returned to their basal concentration under the second temporal window, to stabilize this newly expanded network. Although only a subset of actin regulatory proteins was examined in this study, it is likely that other actin regulators undergo activity-dependent translocations to various spine subcompartments to carry out their function. There are also likely compensatory backup mechanisms to ensure that the actin cytoskeleton grows, or shrinks, as appropriate to the cue or stimulation pattern.

**Table 3 cells-11-00603-t003:** Compartmentalization of actin regulators in dendritic spines.

Compartment	Protein	Notes	Citation
core	Cortactin	Not concentrated in presynaptic terminals. Majority concentrated 100–150 nm away from PSD. Minor population adjacent to PSD.	[107]
	Profilin	Shuttles between core and spine shaft in an activity-dependent manner.	[106,108]
	Drebrin	Stabilizes core filaments by changing mechanical properties of filaments to render them resistant to depolymerization. Number of molecules directly correlates with spine head size.	[109]
Spinoplasm (shell)	Cofilin	Appears completely devoid of spine core. Also found in PSD.	[110]
	Arp2/3 complex	Associates with thin filaments in a “toroidal domain”, between peak cofilin and cortactin concentration. Concentration peaks in spinoplasm, but also sparse localization in PSD. Labels strongly near endosomes.	[69]
	Myosin IIb	Broad distribution, with a preferential localization to spine neck over the spine head. Stabilizes core filaments through crosslinking. Regulates dynamic actin treadmilling through contractility.	[19,54]
PSD	Alpha-actinin	Also associates with the spine apparatus.	[70]
	CaMKII-beta	Enzymatic activity not necessary.	[55,56]

In agreement with the distinct localization of key actin regulators, optical techniques have revealed that dendritic spines contain kinetically distinct pools of actin filaments that presumably differ in function. Star and colleagues first used fluorescence recovery after photobleaching (FRAP) of GFP-tagged actin to show under basal conditions that the spine actin is 85% dynamic and 15% stable [53]. FRAP and other optical techniques have been used to tease apart the distinct roles of the dynamic and stable actin pools. Despite the relative small size of the stable actin pool, it sits at the core of the spine and seems to be critically important for dendritic spine stability [57]. One possibility is that the stable core acts as a base from which the more dynamic peripheral actin network can emanate. This larger dynamic actin pool is enriched at the cell periphery and is critical for expanding spine size [40,102,111] while also organizing and continually remodeling the PSD [98]. Maintaining a balance between these two pools is important for proper spine function and long-term maintenance.

## 6. Conclusions

Actin monomers assemble into highly complex networks of actin filaments, under the direction of many diverse actin regulatory proteins. These actin networks are principal components supporting the elaborate structure of neurons in the brain, including finer structures on the neuron such as pre-synaptic specializations and dendritic spines. A summary can be seen in Table 2. In addition to regulating neuronal structure shape, the actin network plays important roles in synapse function, both pre- and post-synaptically. In the pre-synaptic compartment, a dynamic actin network regulates vesicle release and endocytosis, with both critical for the chemical transmission from one neuron to the next. In the post-synaptic compartment, these signals are received by neurotransmitter receptors whose localization, diffusion, and trafficking are controlled by coupling to an underlying actin network. Moreover, signals received by a synapse often result in the activation, or inhibition, of chemical signaling pathways, many of which result in changes to the underlying actin network.

Since actin is important in determining neuronal structure and structural plasticity, as well as regulating neurotransmission, it is not surprising that many disease-associated causal variants of CNS disorders converge on regulating the actin network, directly by affecting actin regulatory proteins or through indirect mechanisms (a comprehensive review by Muñoz-Lasso and colleagues on cytoskeleton dysfunction in neurological disorders was recently published in this series [112]). Disease mutations in actin-binding proteins manifest in a variety of ways that affect neuron structure and function, including defects in neuronal migration or process extension resulting from defects in the formation and remodeling of actin-rich structures [40,111,113,114], interference with mitochondrial dynamics [115], and perturbations of endocytic recycling [116]. These disruptions contribute to a range of brain disorders including neurodevelopmental disorders, such as autism and intellectual disability; to neurodegenerative disorders, such as sporadic and familial Amyotrophic Lateral Sclerosis (ALS) and Parkinson’s disease; and even to psychiatric disorders, such as schizophrenia and bipolar disorder. The challenge to the field in the coming decades will be to leverage this wealth of genetic information to elucidate the mechanisms that underlie these disorders and to develop methods that target these diseases to prevent or slow their progression.

From the advent of Ramon y Cajal’s first depictions of a neuron and dendritic spines, advancements in light and electron microscopy, live-cell imaging, and genetic tools have greatly advanced our understanding of the neuronal actin cytoskeleton. This has uncovered a wealth of biology on how these beautiful cells are formed and continue to function throughout our lifetime. New genetic insights into the links between actin regulators and CNS disorders position the field to understand the root causes and develop new therapies for treatment.

## Figures and Tables

**Figure 1 cells-11-00603-f001:**
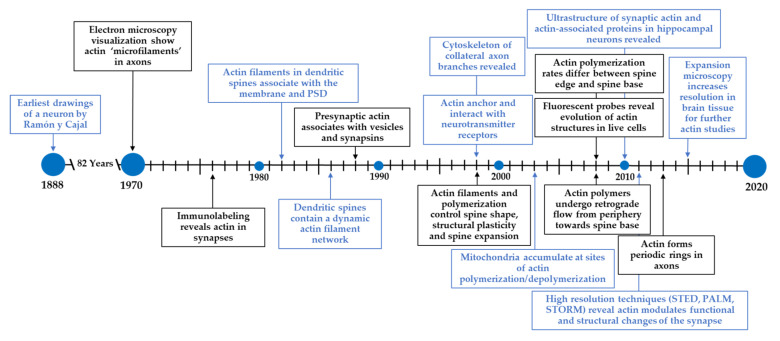
Timeline for the discovery and study of actin at the synapse. Ramón y Cajal pioneered modern-day neuroscience by identifying neurons and dendritic spines. Since then, new technologies have expanded our understanding of neurons and its underlying components. In particular, the field has made tremendous progress in understanding the structure and functions of the actin cytoskeleton in neurons.

**Figure 2 cells-11-00603-f002:**
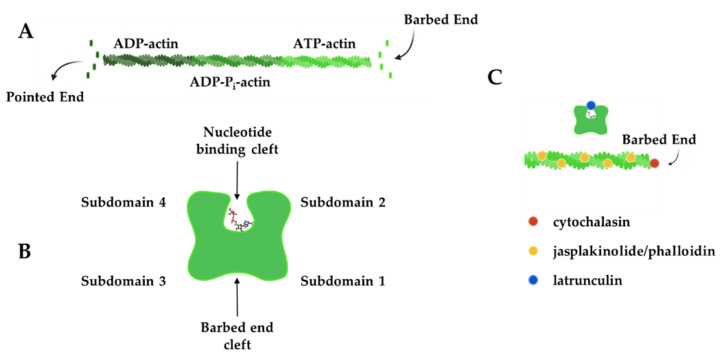
Actin filament dynamics can be modulated by chemical compounds. (**A**) Actin monomers add to the barbed end of actin filaments in an ATP-bound form, while ADP-actin monomers dissociate from the pointed end to create filament treadmilling. (**B**) Actin monomers have four subdomains that are split by a nucleotide-binding cleft. Subdomains 1 and 3 come together to form the barbed-end cleft where many actin-binding proteins interact. (**C**) Chemical compounds can be used to modulate actin dynamics by binding to the barbed end of the filament (cytochalasins), near the nucleotide-binding cleft (latrunculins) of monomers or along the length of actin filaments (jasplakinolide and phalloidin) to exert their effect. Actin filaments in B and C were created and retrieved from https://app.biorender.com/biorender-templates (accessed on 28 January 2022).

**Figure 3 cells-11-00603-f003:**
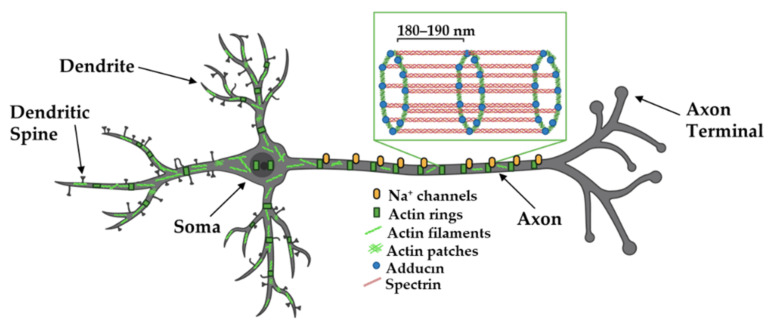
Organization of actin in axons and dendrites. Actin filaments, spectrin, and adducin form ring-like structures around the axon shaft, dendrites, and soma where they are thought to provide structural support and organize axonal Na^+^ channels. Evidence suggests segments of dendrites are either dominated by rings or linear filaments and that neuronal activity causes changes in these structures. Adapted from “Pyramidal Neuron (dendritic spines low)”, by https://BioRender.com (accessed on 26 January 2022) (2020). Created and retrieved from https://app.biorender.com/biorender-templates (accessed on 26 January 2022). Actin-adducin-spectrin image adapted from Xu et al., 2012.

**Figure 4 cells-11-00603-f004:**
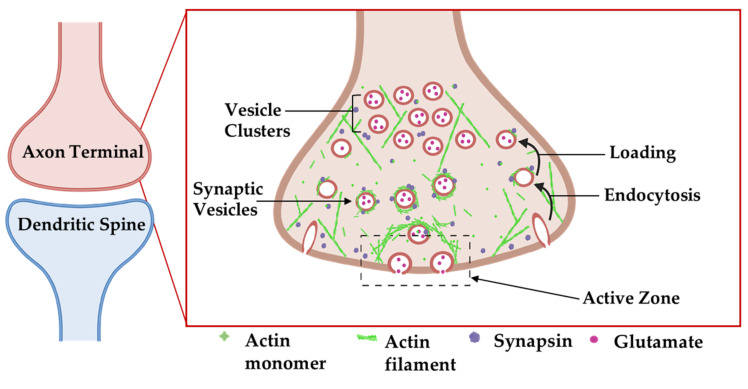
Organization of actin cytoskeleton at the presynaptic axon terminal. Actin filaments form structures presynaptically that affect synapse structure and function. Actin-associated proteins localize to specific areas to carry out their functions. Adapted from “Axonal-dendritic synaptic cleft and Protruding membrane”, by https://BioRender.com (accessed on 26 January 2022) (2020). Created and retrieved from https://app.biorender.com/biorender-templates (accessed on 26 January 2022).

**Figure 5 cells-11-00603-f005:**
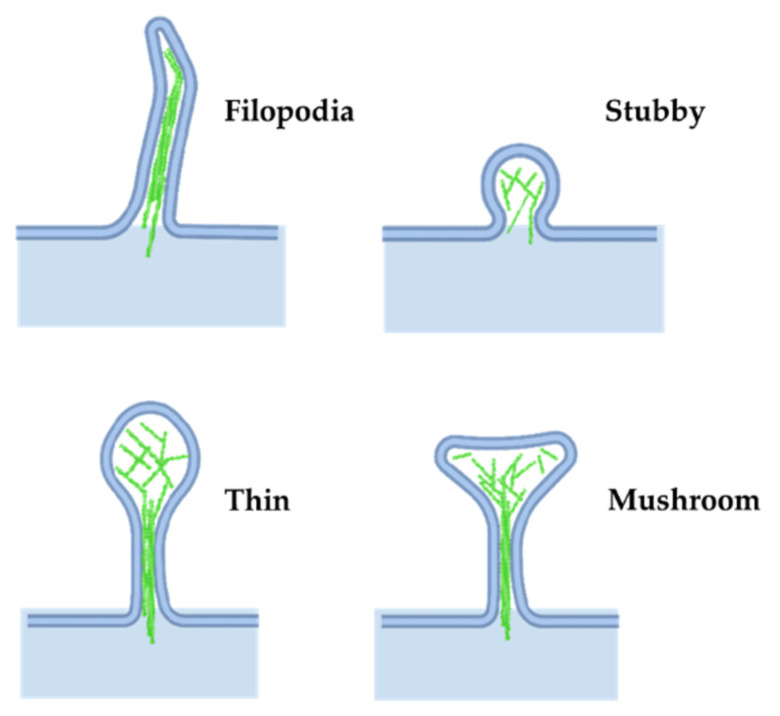
Dendritic spine morphology. Actin networks influence different structures in dendritic spines. Adapted from “Axonal-dendritic synaptic cleft and Protruding membrane”, by https://BioRender.com (accessed on 26 January 2022) (2020). Created and retrieved from https://app.biorender.com/biorender-templates (accessed on 26 January 2022).

**Figure 6 cells-11-00603-f006:**
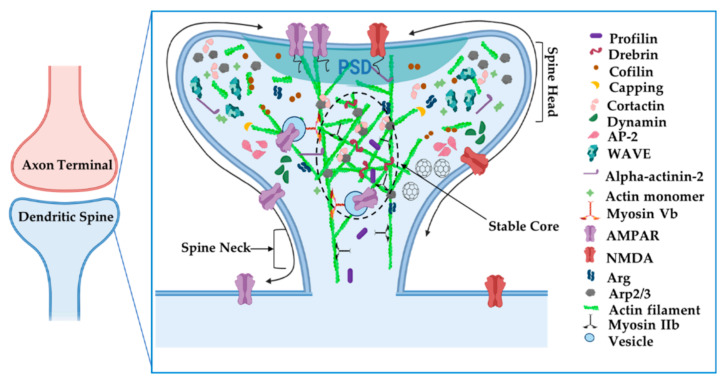
Organization of actin cytoskeleton at the postsynaptic dendritic spine. Actin filaments have specialized compartments, such as the stable actin core, that affect synapse structure and function. Actin-associated proteins localize to specific areas to carry out their functions. Adapted from “Axonal-dendritic synaptic cleft and Protruding membrane”, by https://BioRender.com (accessed on 26 January 2022) (2020). Created and retrieved from https://app.biorender.com/biorender-templates (accessed on 26 January 2022).

**Figure 7 cells-11-00603-f007:**
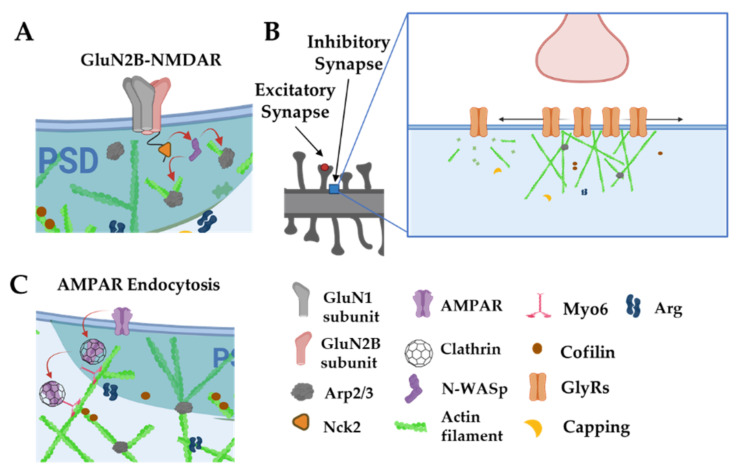
Actin is involved in postsynaptic receptor localization and trafficking. (**A**): The intracellular tails of GluN2B subunits interact with Nck2 to stimulate actin nucleation via N-WASp and the Arp2/3 complex. (**B**): Actin filaments anchor and space out glycine receptors. Disruption of actin filaments leads to smaller, denser clusters of glycine receptors and increases their diffusion out of synaptic zones. (**C**): Myo6 traffics AMPARs away from the PSD through actin filaments. Adapted from “Axonal-dendritic synaptic cleft, Dendritic spine, Post-synaptic membrane and Pre-synaptic membrane”, by https://BioRender.com (accessed on 26 January 2022) (2020). Created and retrieved from https://app.biorender.com/biorender-templates (accessed on 26 January 2022).

**Table 1 cells-11-00603-t001:** The effect of actin inhibitors on filament dynamics.

Name	Chemical Structure *	Origin	Class	Description
Cytochalasins (B PubChem CID 5311281; D PubChem CID 5458428)	B: 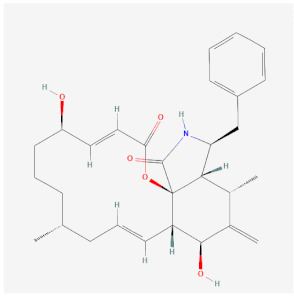 D: 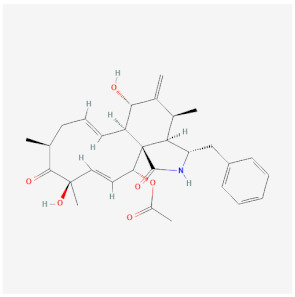	Fungal metabolite	Stabilization/Inhibits further polymerization	Binds to barbed end in a 1 molecule: 1 filament stoichiometry, blocking both assembly and disassembly of monomers. Cell permeable
Jasplakinolide (PubChem CID 9831636)	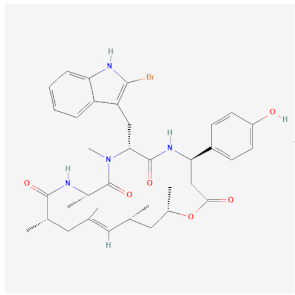	Naturally occurring peptide from *Jaspis johnstoni* sponges	Polymerization	Enhances actin polymerization by lowering the critical concentration of actin. Binds at an interface of three actin monomers. Competes with binding of phalloidin. Cell permeable
Latrunculins (A PubChem CID 445420; B PubChem CID 6436219)	A: 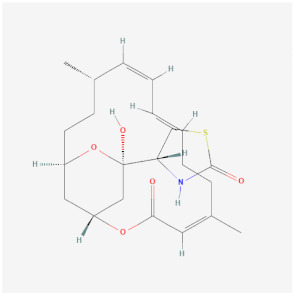 B: 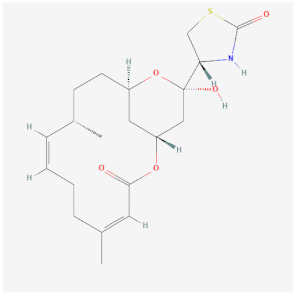	Natural product and toxin produced by certain sponges	Depolymerization	Binds actin monomers near nucleotide-binding cleft with a 1:1 stoichiometry, preventing them from polymerizing. Cell permeable
Phalloidin (PubChem CID 441542_	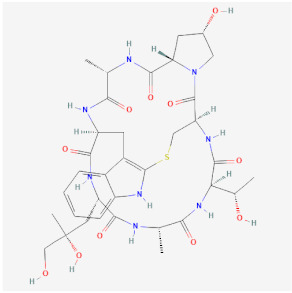	Toxin found in the death cap mushroom *Amanita phalloides*	Stabilization	Binds at the interface of actin subunits preventing depolymerization. Not cell permeable

* All chemical structures were taken from pubchem.ncbi.nlm.nih.gov (accessed on 28 January 2022) [38].

**Table 2 cells-11-00603-t002:** Summary of actin’s role in neuronal compartments.

Compartment	Functions	Reference	Summary
Axon	Axon guidance and terminal arborization	[33,39,40]	Actin polymerization drives neuronal growth cones and pre-synaptic structures
	Regulation of neurotransmitter release	[41]	Depolymerization by latrunculin A enhanced release
	Mechanical support	[34,35,36,37,42]	Actin:spectrin rings support integrity of long-lived neuronal structures
	Organization of sodium channels	[34]	Sodium channels are distributed in a periodic pattern associated with the actin:spectrin cytoskeleton
Dendritic Spines/Excitatory Synapses	Dendritic spine morphogenesis	[20,43,44]	Dynamic changes in actin power formation of dendritic filopodia and their elaboration into dendritic spines
	Structural plasticity, support and stability	[45,46,47,48,49,50,51,52,53,54,55,56,57]	Morphological changes of spines influenced by sensory input and links between actin, its regulators and spine formation, structural plasticity and function
	Regulation of synaptic plasticity	[58,59,60,61,62,63,64]	Changes of the spine neck, spine size and actin-binding proteins are correlated with synaptic strength
	Localization and trafficking of AMPA and NMDA receptors	[65,66,67,68,69,70]	Actin dynamics impact receptor trafficking, recycling and anchoring between synaptic and nonsynaptic zones
	PSD organization	[71]	PSD protein reorganization is driven by changes in actin polymerization
	Microtubule entry into spines	[72]	Actin remodeling promotes microtubule entry into dendritic spines
Presynaptic Terminals	Molecular scaffolding	[73,74,75,76,77]	Actin filaments form a mesh around synaptic vesicles to act as a molecular scaffold
	Regulation of vesicle recycling	[78,74]	Perturbation of actin inhibits recycling and synapsin colocalizes with actin to impact synapse activity
	Regulation of endocytosis timescale	[79,80,81,82]	Actin dynamics impact early stages of endocytosis, all forms of endocytosis (fast, slow, bulk) and replenishment of reserve pools

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
