# Peer review of "Control of Synapse Structure and Function by Actin and Its Regulators"

_cells, 2022, doi:10.3390/cells11040603_

Round 1

Reviewer 1 Report

The manuscript provides a current review of the organisation and distribution of the cytoskeletal protein actin inside neurons and its involvement in various cell biological functions, focusing on pre- and postsynaptic aspects of synaptic transmission. It also attempts to give an overview of the historical development of the tools to study actin. Finally, it presents some information about genetic brain disorders involving actin dys-regulation.  

Given the rapid development in tools and discoveries in neuronal actin research, the chosen topic is timely and interesting. 

However, as is, the article is not very well written, at times confusing, incomplete and superficial.

Criticism/suggestions:

*the timeline (Fig 1) is ineffective and incoherent (e.g. only one reference is named in person, the scale is a bit weird)

*consider giving a clear and more comprehensive description, at least mention, of the technical advances over the years regarding microscopy and labelling. There is no mention of lifeact or other genetically encoded probes, and neither of super-resolution techniques, for instance STORM, PALM and SYED (the biological papers are cited, but not the methods). EM is mentioned but 'platinum replica' or 'rotary shadowing' needs to be explained

*Consider explaining the canonical molecular structure and dynamics of actin monomers and filaments by way of an illustration, also explaining how various 'actin inhibitors' are thought to exert their effects.

*Section 5 on effect of genetic mutations on actin is very weak, consider removing it or making it more worthwhile.           

Reviewer 2 Report

See attached file for comments and suggestions. 

Reviewer 3 Report

In this manuscript, the authors reviewed the literature focusing on the biological functions of actin and its regulator as a central mechanism at neuronal synapses. I really enjoyed reading the review that clearly points out the importance of actin in this field. The review article contains a large amount of detailed information, which is well-structured and easy to understand. The presented information is up-to-date based on references cited by the review. The review would likely be interesting for researchers working in this field

Minor comments:

  1. The authors may want to slightly modify Figure 1 in order to improve the readability. For instance, the text appears slightly out-of-focus (at least the copy I am reviewing). Also, I would suggest changing the gray (for text and boxes) since it does not contrast well with black - maybe select something color-blind-friendly?
  2. The sentence at lines 163-166 is long. I suggest making two sentences out of it. In addition, the end doesn’t make much sense to me. Does something is lacking at the end?
  3. The authors did a great job representing the organization of actin, actin regulators and interacting proteins at synapses (Fig. 3). However, the current size of the figure is problematic since I had difficulties figuring the identity of a specific protein when looking at it as it could be when printed on paper.

Round 2

Reviewer 1 Report

The authors have satisfactorily addressed my comments.

Reviewer 2 Report

The authors have adequately addressed my comments.